# The Key Role of Emotional Repair and Emotional Clarity on Depression among Breast Cancer Survivors

**DOI:** 10.3390/ijerph19084652

**Published:** 2022-04-12

**Authors:** Rocío Guil, Lucia Morales-Sánchez, Paula Ruiz-González, Rocío Gómez-Molinero, Paloma Gil-Olarte

**Affiliations:** 1Department of Psychology, Faculty of Education Sciences, University of Cádiz, 11519 Puerto Real, Spain; rocio.guil@uca.es (R.G.); paula.ruiz@uca.es (P.R.-G.); rocio.gomez@uca.es (R.G.-M.); paloma.gilolarte@uca.es (P.G.-O.); 2Instituto Universitario de Investigación para el Desarrollo Sostenible (INDESS), University of Cádiz, 11406 Jerez de la Frontera, Spain; 3Instituto de Investigación e Innovación Biomédica de Cádiz (INiBICA), University of Cádiz, 11009 Cadiz, Spain

**Keywords:** perceived emotional intelligence, emotional intelligence, emotional repair, emotional clarity, depression, breast cancer, breast cancer survivors, psycho-oncology

## Abstract

Breast cancer is the malignancy with the highest incidence in women worldwide. The empirical evidence is inconsistent with the prevalence of depression among breast cancer survivors (BCS), pointing to emotional competencies as protective factors against affective disorders. However, the mechanisms through which these competencies favor a more adaptive emotional state are unknown. Therefore, this study aims to explore the relationship between the experience of having survived the disease and depression levels in a group of BCS, and the mediating role of Perceived Emotional Intelligence (PEI) in this relation. This was a cross-sectional study with 237 women divided into two groups: 56 BCS and 181 healthy controls who completed the Trait Meta-Mood Scale 24 (TMMS-24) and the Hospital Anxiety and Depression Scale (HADS). Results showed that Survivorship and PEI explained and predicted 37.8% of the variance of depression, corresponding the 11.7% to the direct and/or the indirect effect of the PEI dimensions (Emotional Attention, Emotional Clarity, and Emotional Repair). In conclusion, interventions aimed at promoting an adequate PEI in this population—and in the Psycho-oncology field, in general—with a particular focus on the development of Emotional Clarity and Repair need to be implemented. Limitations and future research lines are discussed.

## 1. Introduction

Despite breast cancer (BC) being one of the most common neoplasms among women, and with the highest incidence in the world [1], the survival rate of women with this disease has undergone an enormous improvement [2]. Notwithstanding, receiving a BC diagnosis continues to be experienced as a traumatic event, due to its short- and long-term impact on the physical, personal, and social levels [3].

On a personal level, the scientific literature shows a high prevalence of affective disorders among women with BC, especially during the first year after diagnosis [4,5]. Depression and anxiety are presented as the two most common symptoms, showing a prevalence of up to 32.2% and 41.9%, respectively [6,7]. Depressive symptoms negatively affect the quality of life and limit the ability to successfully cope with the disease; however, there seem to be differences in depressive levels depending on age [4,8]. In this line, breast cancer survivors (BCS) older than 70 years show higher levels of depression, compared to the general population [8,9]. This supports the studies that confirm how depression is more frequent in older ages, as stated in Girgus et al.’s study [10].

In addition, there is some controversy about the psychological adjustment of women once the disease is over [4]. Some research confirms that they are more vulnerable to experience depressive symptoms than those who have not suffered from the disease [11]. This is usually due to the fear of recurrence or the vital changes derived from the physical, psychological, and social consequences derived from the treatment [12,13]. In addition, these psychological difficulties will often continue to be present even five years after diagnosis [14,15,16]. By contrast, other studies suggest that female survivors show better psychological adjustment and lower levels of depression than healthy women [4,17].

In this regard, numerous investigations highlight the buffering role that other variables can play, such as emotional intelligence (IE), being considered a key protective element against possible emotional disorders. Moreover, it is seen as a facilitator of personal strengthening after facing the diagnosis, and administered treatments [18]. Thus, some research studies showed that one year after diagnosis, patients showed higher levels of resilience [19] and post-traumatic growth [20] than healthy women. A large part of these levels was attributed to their perceived emotional intelligence (PEI).

The relationship between PEI, understood as the perceived ability to attend to, discriminate, and repair emotions [21], and depression is also well documented. In this regard, greater negative emotional states were found in people who perceived little emotional self-efficacy [22,23]. Moreover, the importance of adequate emotional management in adaptation processes has been verified, especially when people face adverse and major changes and situations in their lives [24,25,26], concluding that certain emotional profiles are related to maladaptive behaviors, either in a direct or mediated way.

As stated above, in the oncological field, it has been demonstrated that the difficulty of regulating emotions is associated with greater emotional distress and depression. Likewise, adequate levels of emotional repair act as a protective element in adverse contexts, being related to greater adaptation to the disease [27,28]. In addition, it has been found that older women tend to pay less attention to their emotions [29], which can stimulate the appearance of negative emotional states and depressive symptoms [9,23]. It also confirms the relevant role of age and experience on the development of emotional competencies [30]. In this sense, several studies have examined the levels of PEI in women with BC, showing how the experience of facing the disease increases the ability to perceive oneself as capable of regulating one’s own emotions and those of others [19,31,32,33,34].

Considering the ability of the PEI dimensions to act as risk and/or protective factors against depression, it seems that clarity and emotional repair stand as protective factors against the development of anxious or depressive symptoms [35,36]. Likewise, emotional attention is positively related to negative affective states, dysthymia, and/or depression, and is negatively associated with adequate psychological and social functioning [24,25,37]. Therefore, people who do not pay much attention to their emotions would see their negative emotional states increased if they do not have adequate levels of emotional clarity and repair [38,39]. This mechanism of influence is also confirmed in cancer patients, where people with a higher level of emotional repair show more resilience and quality of life [19]. This evidence confirms the causal relationship between the PEI dimensions, having an adequate perceived capacity to attend to emotions will allow them to be distinguished favorably, stimulating, in turn, the ability to repair their emotional states [33,37,40].

In light of the scientific evidence, the present study aims to understand the mechanisms through which PEI may promote an adequate mental health status in breast cancer survivors (BCS). Concretely, we will explore the relationship between BC survivorship and depression, as well as the possible mediator role of the PEI and its dimensions—attention, clarity, and emotional repair—in this relationship, and the processes by which this influence occurs, controlling for the effect of age. First, it is hypothesized that BCS who have an adequate level of emotional attention (EA), clarity (EC), and repair (ER) will not show pathological levels of depression. Furthermore, as a second hypothesis, we expect to confirm that EA, EC, and ER will explain, either in a direct or mediated way, the absence of depression in women who have experienced BC.

## 2. Materials and Methods

### 2.1. Participants

Participants were 237 women from south Spain divided into two groups: Group 1, composed of 56 BCS (M_age_ = 51.77; *SD* = 8.92); and Group 2, made up of 181 healthy women (M_age_ = 46.87; *SD* = 9.42).

Inclusion criteria to participate in the study for both groups were: (i) to be a woman, (ii) to be older than 18 years, and (iii) to have the ability to speak and read Spanish language. Furthermore, inclusion criteria for Group 1 (BCS) were as follows: (i) BC women with at least one year after their diagnosis or not receiving treatment, and (ii) patients with no history of cancer other than BC. 

Exclusion criteria for both groups were the following: (i) participants who could not read or write, (ii) did not have the adequate cognitive capacity to understand the instruments, (iii) presence of psychiatric or neurological disorders such as dementia or psychosis, (iv) receiving psychiatric or psychological treatment due to the presence of a mental disorder, (v) being under the influence of psychoactive medication, and (vi) having any concomitant disabling pathology.

The inclusion criteria to participate in the study for both groups were: (1) to be a woman; and (2) to be older than 18 years. Moreover, for the group of breast cancer women, other inclusion criteria were applied: (1) to have a breast cancer pathology. In addition, those women who (1) could not read or write, (2) did not have the adequate cognitive capacity to understand the instruments, and (3) were under the influence of psychoactive medication, were excluded.

Regarding the recruitment process, the conscription of women from Group 1 (BCS) was carried out through the nursing staff of the oncology units of the three hospitals in the province of Cádiz (Spain), in coordination with the members of the research team of the project ‘PIN-0109-2018’. They explained the possibility to participate in the Project to those women who met the eligibility criteria. For those women who agreed to participate, we cite them to a personal interview to explain the details of the Project and their possibility to leave their participation at any moment, as well as to address the online questionnaire. Furthermore, the selection of women from Group 2 (healthy women) was carried out through electronic advertisements and posters in the Hospitals and Health Centers of the province. Those women who were interested to participate in the study contacted us and we also cited them as in the case of Group 1.

Participation was voluntary and all subjects previously signed an informed consent. In this line, all participants completed the questionnaires through an online survey by digital devices and under the supervision of research assistants. Volunteers received no financial compensation for the participation in the study.

### 2.2. Ethical and Legal Considerations

The study was approved by the Ethical Committee of Research (CEI/CEIm) of the Health Ministry of Andalusian (code 82/2018), and it is part of the project ‘e-Health para la Promoción de la Salud y la Calidad de Vida en Oncología Mamaria’ (PIN-0109-2018).

The study was carried out in the three hospitals of the province of Cádiz, a province of Andalusian (Spain): The University Hospital Puerta del Mar, the University Hospital of Puerto Real, and the University Hospital of Jerez de la Frontera.

Data were collected according to the general ethical research principles of the American Psychological Association (APA). Moreover, the research was executed under the Spanish and European regulations for the research and privacy protection of personal data. In addition, under the Circular 15/2001 of the AEM (Medicines Act); the Royal Decree 711/2002; the recommendations of the Good Clinical Practice of the EEC (document 111/3976/88, July 1990); and the requirements expressed in the Helsinki Declaration.

### 2.3. Instruments

Depression was assessed by the use of The Hospital Anxiety and Depression Scale (HADS; Zigmond and Snaith [41]; adapted by Terol et al. [42]. This 14-item self-report scale assesses emotional distress through 2 sub-scales of 7 items each: anxiety and depression. Only the depression dimension was used. The responses were collected through a Likert-type scale from 0 to 3 points. A score equal to or less than 7 indicates that there are no depressive symptoms, between 8 and 10 reflect the presence of symptoms, and greater than or equal to 11 is indicative of a diagnosis of depression. The authors of the scale found that Cronbach’s alpha for the depression subscale was 0.71 [42].

Perceived Emotional Intelligence, was evaluated by The Trait Meta-Mood-Scale 24 (TMMS-24; Salovey et al. [21]; adapted by Fernández-Berrocal et al. [43]. This 24-item questionnaire assesses people’s perception of their emotional abilities. It consists of 3 dimensions that assess the degree to which subjects think about their emotions (emotional attention), understand their emotional states (emotional clarity), and can regulate them (emotional repair). The response format is Likert-type with a range from 1 to 5. For women, values below 24 represent little attention, between 25 and 35, adequate levels of it, and above 36, excessive attention to emotional states. Regarding the emotional clarity and repair dimensions, values below 23 represent low levels, between 24 and 34 pose adequate status, and above 35 indicate excellent emotional clarity and repair levels. The authors of the instrument found that the reliability coefficient, Cronbach’s alpha, was 0.90 for emotional attention, 0.90 for emotional clarity, and 0.86 for emotional repair [43].

To control the effect of age, a self-elaborated sociodemographic questionnaire was designed, which included the age of the patients. Moreover, this socio-demographic questionnaire included civil status, employment situation, level of education, having children or not, and the number of children.

### 2.4. Statistical Analyses

Preliminary analyses were carried out to determine the descriptive statistics data of the sample, differentiating between both groups of women, and the total sample. In addition, these analyses were conducted to obtain the reliability of the instruments; to gather the Pearson’s bivariate correlations to observe the possible associations between the variables; and to perform the analysis of variance (ANOVA) to determine the differences between the means of the group. All the analyses were carried out using the IBM SPSS v.22 (IBM Corp, Chicago, IL, USA).

Regarding the effect size of the ANOVA, it was calculated through Cohen’s *d*: values between 0.20 and 0.30 were considered small effects; those between around 0.50 and 0.80 were medium effects; and more than 0.80, were counted as large effects. Concerning the effect size of Pearson correlations, it was calculated based on the correlation coefficients where values were deemed as follows: values between 0.10 and 0.30, small effects; values between 0.30 and 0.50, moderate effects; and ≥0.50, high effects [44].

Finally, a Serial Multiple Mediation Analysis (specifically, Model 6) was carried out through the PROCESS tool of SPSS [45]. Accordingly, a bootstrapping resampling method (10,000 simulations) was executed to construct the 95% confidence intervals.

## 3. Results

### 3.1. Socio-Demographic Characteristics of the Participants

Regarding the characteristics of the participants, the women who composed both groups presented the following aspects:

Group 1 (breast cancer women): Civil status (69.6% married, 10.7% widowed, 7.1% civil, 8.9% single, and 3.6% widowed); Employment situation (12.5% active, 19.6% housewife, 17.9% sick leave, 17.9% unemployed, and 32.1% pensioner). Level of education (35.7% university studies, 17.9% professional training, 12.5% secondary studies, 30.4% primary studies, and 3.6% no studies). On the other hand, 78.8% have children, while 21.4% have no children. Of those women with children, the mean number of children is 1.67 (M = 1.67; *SD* = 1.14).

Group 2 (healthy women): Civil status (50.3% married, 13.8% widowed, 8.8% civil partner, 25.4% single, and 1.7% widowed). Employment situation (75.7% active, 5.0% housewife, 1.1% sick leave, 10.5% unemployed, and 7.7% pensioner). Level of education (73.5% university studies, 11.6% professional training, 13.8% secondary studies, and 1.1% primary studies). Of the participants, 62.4% have children, while 37.6% do not. Of those who have children, the mean number of children is 1.15 (M = 1.15; *SD* = 1.06).

### 3.2. Descriptive Statistics of the Variables Included in the Study

Table 1 shows the descriptive statistics data regarding the variables included in the study for all participants, and for the two sample groups separately. As may be observed, BCS presented higher levels of depression than control subjects. However, taking into account the interpretation of the scale, none of the groups showed psychopathological depressive symptoms. Regarding PEI, results indicated that women in both groups exhibited adequated values of EA, EC, and ER.

To check if the differences found among the values of both groups were statistically significant, means contrasts were carried out through analysis of variance (ANOVA) with all the study variables. The results indicated that BCS showed significantly higher levels of depression (*F* (1, 235) = 81.48; *p* < 0.001), lower EC values (*F* (1, 235) = 10.11; *p* < 0.05), and older ages (*F* (1, 235) = 11.87; *p* < 0.001), than those of the control group. However, no statistically significant differences were encountered regarding EA (*F* (1, 235) = 1.19; *p* > 0.05) and ER values (*F* (1, 235) = 0.01; *p* > 0.05). Moreover, Cohen’s *d* results demonstrated a strong effect size for depression (*d* = 1.02), medium effect for EC (*d* = −0.48), and for age (*d* = 0.53).

### 3.3. Correlational Analyses

Table 2 includes the bivariate correlations of all the outcome variables. Considering the value of the Pearson correlation coefficients, depression showed a large positive association with BC (*r* = 0.51; *p* < 0.001). Moreover, a negatively relation, although small, with EC (*r* = −0.28; *p* < 0.001), and moderate with ER (*r* = −0.31; *p* < 0.001). Moreover, other statistically significant and negative relationships, although small, were found between BC and EC (*r* = −0.20; *p* < 0.001), moderate and positive between EA and EC (*r* = 0.35, *p* < 0.001), and also strong between EC and ER (*r* = 0.53; *p* < 0.001). Regarding age, results also revealed significant positive and small correlations with BC (*r* = 0.22; *p* < 0.001) and depression (*r* = 0.17; *p* < 0.001), as well as negative with EA (*r* = −0.16; *p* < 0.001).

### 3.4. Mediational Analysis

Finally, to explore the direct and/or indirect-mediated effects of the PEI dimensions in the relationship between BC survivorship and depression, a Serial Multiple Mediation Analysis was performed through Model 6 of the Macro PROCESS [46]. Thus, depression was included as the dependent variable (Y), BC as the independent variable (X), and the three dimensions of the PEI –EA (M_1_), EC (M_2_), and ER (M_3_)— as mediators. Furthermore, Age was included in this model as a covariate due to the results found in previous analyses.

The main direct and mediated effects are reflected in Figure 1. The global direct effect indicated that all the variables included in the mediation analysis explained 38% of the variance of depression (*R*^2^ = 0.38; *c*′: *β* = 2.72; BootSE = 0.31; *p* < 0.001; BootCI 95% (2.12, 3.33)), while the total effect showed that BC and age explained the 26.1% of it (*R*^2^ = 0.26; *c: β* = 2.75; BootSE = 0.32; *p* < 0.001; BootCI 95% (2.12, 3.39)).

Furthermore, the model indicated that BC predicts lower levels of EC (a_2_ = −3.05; Boot*SE* = 0.96; *p* < 0.05; BootCI 95% (−4.94, −1.15)), but higher ER (a_3_ = 1.99; BootSE = 0.80; *p* < 0.05; BootCI 95% (0.40, 3.59)). Other direct effects showed that women who pay more attention to emotions have greater EC (d_21_ = 0.34; BootSE = 0.06; *p* < 0.001; BootCI 95% (0.22, 0.46)). Furthermore, higher EC increases ER states in all women of the sample (d_32_ = 0.53; BootSE = 0.05; *p* < 0.001; BootCI 95% (0.43, 0.64)). Finally, the direct effects indicated that high levels of EA increase depression values (b_1_ = 0.05; BootSE = 0.02; *p* < 0.05; BootCI 95% (0.01, 0.09)). However, it is observed that high levels of ER decrease depressive symptoms in our entire sample (b_3_ = −0.11; BootSE = 0.02; *p* < 0.001; BootCI 95% (−0.16, −0.06)).

Considering the mediating influence of the PEI dimensions, we found two significant indirect effects. On one hand, the Indirect Effect 3 indicated that BC leads to higher levels of ER, assuming a reduction in levels of depression (Ind_3_ = −0.22; Boot*SE* = 0.11; BootCI 95% (−0.47, −0.02)). On the other hand, the Indirect Effect 6 reported that BC reduces the EC, which decreases levels of ER, increasing in turn depressive symptoms (Ind_6_ = 0.18; Boot*SE* = 0.09; BootCI 95% (0.03, 0.41)). Contrast analyzes between indirect effects revealed that indirect effect 6 (Ind_6_) showed the most statistical weight (C14 = −0.41; BootSE = 0.17; BootCI 95% (−0.77, −0.11)). This points out that the levels of depression in BCS can be increased, being explained by the serial effect that the illness exerts on EC and this, in turn, on ER.

## 4. Discussion

After reviewing the literature, there is some controversy about the existence of depressive symptoms in BC patients [4,5,6,7,8,9,11,12,13,14,15,16,17]. Although some emotional competencies, such as EI, have been pointed out as a protective factor against mental health, the present research tries to show the precise mechanisms through which PEI influences the incidence of depression among BCS.

Concerning the descriptive statistics and confirming our initial hypothesis, we observe that the total sample does not present symptoms of depression, while also showing adequate levels of PEI dimensions. Even though certain studies reveal the presence of anxiety-depressive symptoms in women who have experienced a BC diagnosis [7,14,16], the obtained results are in line with research that points out the good psychological adjustment of BCS once the cancer is over [4,18,20]. In addition, the PEI levels show that, despite having experienced a traumatic event such as a cancer diagnosis, patients show an adequate capacity to attend to, perceive, and regulate their emotions [19,32,33].

Taking into account the results of the analysis of variance, it was found that although BCS do not present clinically significant symptoms of depression, they show a higher vulnerability to experiencing depressive symptoms than women without the disease [10]. This greater predisposition may be due to the fear of recurrence, as well as due to once the treatments finish, BC women are obliged to readjust their life in multiple facets in consequence of the physical, psychological, and social sequels derived from the illness [12,13].

Moreover, in contrast to the study of Cejudo et al. [31], in which differences in the PEI dimensions between BCS and healthy controls were not found, this research partially shows different results. Although there are no significant differences regarding the attention paid to emotions, or the ability to regulate negative emotional states between both groups, BCS show greater difficulty in discriminating and understanding their emotional states. Namely, it seems that having suffered from the disease and the impact associated with it has different repercussions on each of the PEI dimensions [19,32].

Regarding the correlations, a link between survivorship and depression was found. Accordingly, the present investigation supports the studies that show that those patients who have faced a BC diagnosis more frequently experience negative affective states [5,16]. Furthermore, the negative relationship obtained between BC and EC indicates that female survivors seem to perceive themselves as less self-effective in understanding their emotions. This may be because the cancer diagnosis may imply the experimentation of very varied emotions by the patients, which make them overwhelmed, and make it difficult for them to adequately identify and understand their moods [46].

With respect to depression, the results also show negative associations with EC and ER [22]. In this regard, those women who have a limited perceived capacity in distinguishing emotional states and/or in regulating their emotions will be more likely to experience depressive symptoms [32,36,37]. In addition, it was found that older women will manifest higher negative emotional states [8,9] and will tend to pay less attention to their emotions [29]. In this line, following further exhaustive analyses, it may be expected that the combined effect of advanced age and low EA may have a significant impact on the increase in depressive symptomatology, independently of the BC diagnosis.

Additionally, the positive relationships found between the PEI dimensions suggest that paying adequate attention to emotions is positively associated with the clarity with which they are perceived and this, in turn, with the adequate capacity to regulate and repair negative emotional states that interfere in the adaptative functioning [19,25,33,40].

Considering the mediation model, the second hypothesis of the study is confirmed. In line with the previous findings, the direct effects showed that BC survivorship predicts the appearance of negative mood, anhedonia, abulia, or apathy, among other depressive symptoms [2,15]. Likewise, it was also found that even though women who have suffered from this oncological disease will perceive a lower capacity to understand and discriminate between different emotions, they will adequately regulate their negative emotional states [19,33,34].

Moreover, in line with previous research, it was found that depression is closely associated with high standards of EA, that is a characteristic of persistent rumination of affective and mood disorders [24,37]. This notwithstanding, levels of depression are lower in those women who trust in their ability to regulate and repair negative affective states. It will lead, in turn, to other positive health outcomes, such as resilience and higher satisfaction with life [19,38,39].

Concerning PEI, other direct effects are linked EA to EC, and this, in turn, to ER. In this sense, those women who adequately attend to their emotions will tend to discriminate and understand them more effectively and repair them successfully [19,34]. Finally, it was found that when age increases, levels of EA decrease, suggesting that life experiences can be decisive in how people handle their emotions [19,30].

About the indirect effects, two fundamental routes were observed. On one hand, it was found that BCS are more vulnerable to experiencing depressive symptoms if they perceive themselves to have low self-efficacy in understanding their emotions [23]. In turn, it would negatively impact their ability to effectively repair negative emotions. Therefore, poor EC would act as a risk factor, facilitating the development of psychopathological comorbidities. On the other hand, and in line with research that highlights the buffering role of EI [18,20], it was also found that ER can reverse this process acting as a protective factor [19,25]. Thus, BCS will experience fewer depressive symptoms if they perceive an adequate capacity to repair negative emotional states [27,28].

Regarding the strengths of the study, results can explain the existing controversy about the presence or absence of depression in BCS and how a personal resource such as PEI can act as both protector or risk factor, depending on which dimensions of PEI (EA, EC, or ER) show greater presence in the sample. In this line, we will find cases in which the different dimensions of the PEI can act as a protective factor against the development of affective disorders, reducing their incidence; or, on the contrary, as a risk factor, increasing vulnerability to experiencing negative emotional states and making adaptation difficult after the end of treatment [22].

Therefore, the findings of the research contribute to the improvement of the health and quality of life of breast cancer survivors and, in turn, of the psycho-oncology field. In this sense, the results can be used to the development of intervention programs aimed at reducing or preventing depression in women with breast cancer. Moreover, it would let health professionals know the importance of training emotional intelligence in this population. Finally, the study highlights the need to implement policies that include emotional intelligence as a psycho-social resource that can be promoted in clinical and no clinical populations, and it would impact the health standards of the societies.

Despite the new findings, this research is not without limitations. Firstly, it is a cross-sectional study since the levels of the variables recorded before or even during the disease are not known. In addition, the use of measures based on self-reports may mean that the responses were biased due to social desirability or the halo effect, among others. Moreover, a bigger sample of BCS could improve the obtained results. Furthermore, other clinical variables may influence the results, such as stage of disease or type of treatment. As future lines of research, we propose conducting a moderation analysis to find out what levels of emotional attention, clarity, and repair are adequate to avoid the onset of depression. In addition, performing these analyses in women who are receiving treatment, as well as in other types of oncological diseases, would allow us to check whether these results are generalizable to other populations.

In addition, taking into account other sociodemographic variables and the anxiety dimension of the scale would increase the value of the article. Moreover, the use of other clinical and paraclinical evaluations would complement the extracted data and achieve more valuable results.

## 5. Conclusions

This mediation model explains the existing controversy about the presence or absence of depression in BCS. As can be seen, depending on the patients’ perceived emotional intelligence, we will find cases in which the different dimensions of the PEI can act as a protective, or as a risk factor, against the development of affective disorders. 

In this line, it is important to emphasize that is not only necessary to design intervention programs focused on the training on the emotional skills of attention or emotional repair, but it will be essential to previously know the clarity with which emotions are perceived to avoid unwanted effects. Understanding emotions is a determining factor to facilitate learning ability, and to let BCS manage their well-being and their lives once the disease is over [26]. Similarly, it would contribute to a lower prevalence of depressive disorders in both clinical and non-clinical populations, thus translating into better population mental health.

## Figures and Tables

**Figure 1 ijerph-19-04652-f001:**
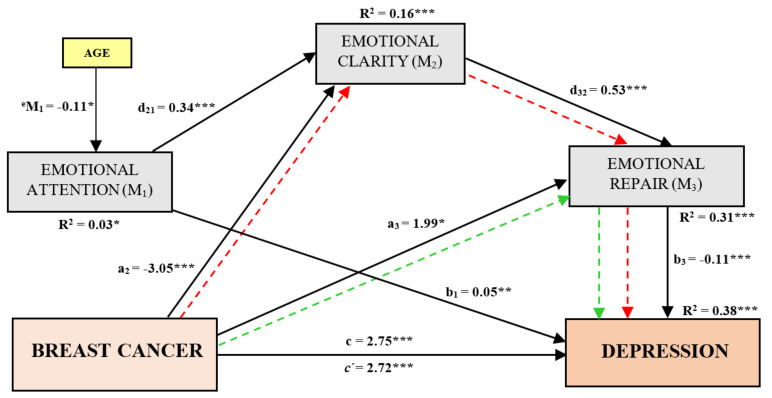
Direct and indirect effects for the proposed Serial Multiple Mediation Model. Notes: The variables used in the mediation model were the following: X = Independent Variable: breast cancer; Y = Dependent Variable: depression; M_1_ =Mediator 1: emotional attention; M_2_ = Mediator 2: emotional clarity; M_3_ = Mediator 3: emotional repair; Covariate = age. Effects: a_2_ = Direct effect of BC on EC; a_3_ = Direct effect of BC on ER; b_1_ = Direct effect of EA on depression; b_3_ = Direct effect of ER on depression; d_21_ = Direct effect of EA on EC; d_32_ = Direct effect of EC on ER; ^e^M_1_ = Direct effect of Age on EA; *c*′ = Direct effect of BC on depression; c = Total effect of BC on depression; R^2^ = R square. * = *p* < 0.05; ** = *p* < 0.01; *** = *p* < 0.001. 

 Direct effects. 

 Indirect Pathway 3 (a_1_b_1_) = Indirect effect of BC on depression through ER only. 
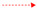
 Indirect Pathway 6 (a_2_d_32_b_3_) = Indirect effect of BC on depression through EC and ER in serial.

**Table 1 ijerph-19-04652-t001:** Descriptive statistics of all the variables included in the study for the total sample, and both groups separately: Group 1 (breast cancer survivors), and Group 2 (healthy women).

			Total Sample			Group 1				Group 2			
	*N*	α	Range	M	SD	*N*	Range	M	SD	*N*	Range	M	SD
AGE	237		29–71	48.03	9.51	56	29–67	51.77	8.92	181	29–71	46.87	9.42
DEPRESSION	237	0.90	0–16	1.82	2.38	56	0–16	3.98	3.78	181	0–5	1.15	1.06
EA	237	0.88	8–40	24.72	6.76	56	8–40	24.50	7.04	181	8–39	24.93	6.67
EC	237	0.92	11–40	29.57	6.65	56	11–40	27.14	7.18	181	11–40	30.31	6.31
ER	237	0.90	11–40	30.24	6.01	56	14–40	30.32	7.16	181	11–40	30.22	5.63

Notes. EA (emotional attention); EC (emotional clarity); ER (emotional repair); Group 1 (breast cancer survivors); Group 2 (healthy women).

**Table 2 ijerph-19-04652-t002:** Bivariate Pearson correlations between all the study variables.

	AGE	BC	EA	EC	ER	DEPRESSION
AGE	-					
BC	0.22 **	-				
EA	−0.16 *	−0.07	-			
EC	−0.02	−0.20 **	0.35 **	-		
ER	−0.06	0.01	0.11	0.53 **	-	
DEPRESSION	0.17 **	0.51 **	0.04	−0.28 **	−0.31 **	-

Notes. BC (breast cancer); EA (emotional attention); EC (emotional clarity); ER (emotional repair); ** *p* < 0.01; * *p* < 0.05.

## Data Availability

Datasets for the current study are available from the corresponding author upon reasonable request.

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
