# Peer review of "The Key Role of Emotional Repair and Emotional Clarity on Depression among Breast Cancer Survivors"

_ijerph, 2022, doi:10.3390/ijerph19084652_

Round 1

Reviewer 1 Report

Dear Autors, 

I'm wondering you decided to use ANOVA test?

Wouldn't it be better to compare group 1 (BCS) and group 2 (healthy women) with t-student test? 

Author Response

Response to reviewer 1 comments

Point 1:

Dear Authors, 

I'm wondering you decided to use ANOVA test?

Wouldn't it be better to compare group 1 (BCS) and group 2 (healthy women) with t-student test? 

Response to point 1:

Dear reviewer,

Firstly, thank you for your kindly recognition and your useful comment. The response is the following:

Regarding your question, yes, a one-factor ANOVA test has been used. The t-student test could also have been used. However, the ANOVA test was chosen to compare whether the differences between groups in the study variables were significant with respect to one factor (having breast cancer or not).

Both tests are indeed used to compare the means of two or more groups. And, specifically, the t-Student is used to compare two groups. On the other hand, the ANOVA test is used to compare the means of two or more groups. Therefore, the t-Student analysis would also have been appropriate. Nevertheless, the data obtained from the ANOVA, comparing the group of women with cancer and without cancer, also reflects the results we wanted to obtain: whether the comparisons of means of the variables between the two groups were significant or not regarding the study variables: Depression, Perceived Emotional Intelligence (Emotional Attention, Emotional Clarity, and Age).

On the other hand, although the one-factor ANOVA test does not give us which group is the one that presents higher or lower levels of significant differences, we can verify this fact due to the descriptive analyses carried out with all the variables in the study, as shown in Table 1. This table reflects the descriptive data (mean and standard deviation) of each of the variables for the group of women with breast cancer, for the group without breast cancer, as well as for the total sample.

Nevertheless, we thank you for your appreciation and we will consider the possibility of performing a t-Student analysis on a future occasion when we are going to compare only two groups.

Reviewer 2 Report

Dear Authors, 

you have presented interesting manuscript, but I have some comments about it. 

Introduction section is quite long, could you make it shorter. 

Section Materials nad Methods 
- add information when and where the study was done
- add inclusion and exclusion criteria for both groups   
- could you explain what does it mean that participants completed the survey through digital devices?
- to the instruments add information about the internal consistency of the questionnaires - Cronbach’s alpha
- in the statistical analysis there is p-value missed 

In section Results there is no information about characteristic of participants.

In Discussion in the first statement line 224-225 you have point out there are some controversy in literature but as a Reader I can not chcek in what studies - please add references. 

Strength and limitations of your study should be as last paragraph in section Discussion. 

Could you explain is this resaerch Guil R. et al. Breast cancer and 397 resilience: The controversial role of Perceived Emotional Intelligence. Front. Psychol. 2020a, 11, 3604. DOI: 398 10.3389/fpsyg.2020.595713 connetcted some how with the manuscript presented in MDPI? 

Please chcek your references for example position 26 - in the journal there is year 2021 but you have wrote 2020 - check it carefully. 

Author Response

Response to reviewer 2 comments

Dear Authors, 

you have presented interesting manuscript, but I have some comments about it. 

Point 1:

Introduction section is quite long, could you make it shorter. 

Response to point 1:

Dear reviewer, thank you for your suggestion. However, it is difficult to make it shorter because during the section we try to justify our research, and eliminating some phrases, words, or references, could change the sense of our introduction.

Therefore, if we reduce it, maybe we will not be able to explain the state of the question of our investigation.

I hope it would be not a problem. Thank you in advance for your understanding and your comments.

Point 2:

Section Materials and Methods 

  • Add information when and where the study was done.

Response to point 2.1.:

Dear reviewer, this information has been added in the lines 145-148:

“Besides, the study started in December 2019 and it is still active until December 2022. It is carried out in the three hospitals of the province of Cádiz, a province of Andalusian (Spain): The University Hospital Puerta del Mar, the University Hospital of Puerto Real, and the University Hospital of Jerez de la Frontera.”

  • Add inclusion and exclusion criteria for both groups  

Response to point 2.2.:

Dear reviewer, the information about inclusion and exclusion criteria has been added in lines 108-118:

Inclusion criteria to participate in the study for both groups were: (i) to be a woman, (ii) to be older than 18 years, and (iii) to have the ability to speak and read the Spanish language. Furthermore, inclusion criteria for Group 1 (BCS) were as follows: (i) were as follows: (i) BC women with at least one year after their diagnosis or not receiving treatment, and (ii) patients with no history of cancer other than BC.

Exclusion criteria for both groups were the following: (i) participants who could not read or write, (ii) did not have the adequate cognitive capacity to understand the instruments, (iii) presence of psychiatric or neurological disorders such as dementia or psychosis, (iv) to receive psychiatric or psychological treatment due to the presence of a mental disorder, (v) being under the influence of psychoactive medication, and (vi) having any concomitant disabling pathology.”

  • Could you explain what does it mean that participants completed the survey through digital devices?

Response to point 2.3.:

With regard to the survey implementation through digital devices, we meant that we elaborated an online questionnaire which included all the instruments used in the study, as well as other psycho-social variables related with health and quality of life. Hence, to clarify this aspect, the following information has been added in lines 136-139:

Participation was voluntary and all subjects signed previously an informed consent. In this line, all participants completed the questionnaires through an online survey by digital devices and under the supervision of research assistants. Volunteers received no financial compensation for participation in the study”.

  • To the instruments add information about the internal consistency of the questionnaires - Cronbach’s alpha

Response to point 2.4.:

Dear reviewer, thank you for your recommendation. The information about the internal consistency of the questionnaires has been added, as you suggested.

Please, see line 163: “The authors of the scale found that Cronbach's alpha for the depression subscale was 0.71.” and lines 174-175: “The authors of the instrument found that the reliability coefficient, Cronbach’s alpha, was 0.90 for Emotional Attention, 0.90 for Emotional Clarity, and 0.86 for Emotional Repair.

  • In the statistical analysis there is p-value missed 

Response to point 2.5.:

Dear reviewer, thank you for your comment, and sorry for the mistake. We have changed the p-values in the notes of Table 2 and added them correctly, as stated in the correlational analysis.

Point 3:

In section Results there is no information about characteristic of participants.

Response to point 3:

Dear reviewer,

Thank you for your suggestion. In the section results the characteristic of the participants has been added. Namely, the following information has been included (please, see lines 199-212):

Regarding the characteristics of the participants, the women who composed both groups presented the following aspects:

Group 1 (Breast Cancer Women): Civil status (69.6% married, 10.7% widowed, 7.1% civil, 8.9% single, 3.6% widowed); Employment situation (12.5% active, 19.6% housewife, 17.9% sick leave, 17.9% unemployed, 32.1% pensioner). Level of education (35.7% university studies, 17.9% professional training, 12.5% secondary studies, 30.4% primary studies, 3.6% no studies). On the other hand, 78.8% have children, while 21.4% have no children. Of those women with children, the mean number of children is 1.67 (M = 1.67; SD = 1.14).

Group 2 (healthy women): Civil status (50.3% married, 13.8% widowed, 8.8% civil partner, 25.4% single, 1.7% widowed). Employment situation (75.7% active, 5.0% housewife, 1.1% sick leave, 10.5% unemployed, 7.7% pensioner). Level of education (73.5% university studies, 11.6% professional training, 13.8% secondary studies, 1.1% primary studies). Of the participants, 62.4% have children, while 37.6% do not. Of those who have children, the mean number of children is 1.15 (M = 1.15; SD = 1.06).

Point 4:

In Discussion

  • In the first statement line 224-225 you have point out there are some controversy in literature but as a Reader I can not chcek in what studies - please add references. 

Response to point 4.1.:

Dear reviewer, thank you for your comment, and sorry for the mistake. References have been added in the paper. Please, see line 289.

  • Strength and limitations of your study should be as last paragraph in section Discussion. 

Response to point 4.2.:

Dear reviewer, following your suggestions and comments, strengths and limitations have been added as the last paragraphs in section Discussion. The following information has been included in the paper (please, see lines 362-392):

Regarding the strengths of the study, results can explain the existing controversy about the presence or absence of depression in BCS and how a personal resource as PEI can act as both protector or risk factor, depending on what dimensions of PEI (EA, EC, or ER) show greater presence in the sample. In this line, we will find cases in which the different dimensions of the PEI can act as a protective factor against the development of affective disorders, reducing their incidence; or, on the contrary, as a risk factor, increasing vulnerability to experiencing negative emotional states and making difficult the adaptation after the end of treatment [22].

Therefore, the findings of the research contribute to the improvement of the health and quality of life of breast cancer survivors and, in turn, of the psycho-oncology field. In this sense, the results can be used to the development of intervention programs aimed at reducing or preventing depression in women with breast cancer. Moreover, it would let health professionals know the importance of training emotional intelligence in this population. Finally, the study highlights the need to implement policies that include emotional intelligence as a psycho-social resource that can be promoted in clinical and no clinical populations, and it would impact the health standards of the societies.

Despite the new findings, this research is not without limitations. Firstly, it is a cross-sectional study since the levels of the variables recorded before or even during the disease are not known. In addition, the use of measures based on self-reports may mean that the responses were biased due to social desirability or the halo effect, among others. Also, a bigger sample of BCS could improve the obtained results. Furthermore, other clinical variables may influence the results such as stage of disease or type of treatment. As future lines of research, we propose conducting a moderation analysis to find out what levels of emotional attention, clarity, and repair are adequate to avoid the onset of depression. In addition, performing these analyses in women who are receiving treatment, as well as in other types of oncological diseases, would allow us to check whether these results are generalizable to other populations.

Also, taking into account other sociodemographic variables and the anxiety dimension of the scale would increase the value of the article. Moreover, the use of other clinical and paraclinical evaluations would complement the extracted data and achieve more valuable results.”

  • Could you explain is this resaerch Guil R. et al. Breast cancer and 397 resilience: The controversial role of Perceived Emotional Intelligence. Front. Psychol. 2020a, 11, 3604. DOI: 398 10.3389/fpsyg.2020.595713 connetcted some how with the manuscript presented in MDPI? 

Response to point 4.3.:

Yes, of course. The research yielded in ‘Guil R. et al. Breast cancer and resilience: The controversial role of Perceived Emotional Intelligence. Front. Psychol. 2020, 11, 3604. DOI: 398 10.3389/fpsyg.2020.595713’ is connected with the study presented in MDPI because both are included in the global project that our research team 'HUM-843 Emotional Intelligence' carry out: E-Health para la Promoción de la Salud y la Calidad de Vida en Oncología Mamaria (PIN-0109-2018).

This project aims to explore numerous psycho-social and emotional variables in the context of breast cancer. Concretely, one of the most important variables to us is 'Emotional Intelligence', and we pretend to demonstrate the role that it has on other variables such as psychological well-being, post-traumatic growth, etc., as it has been done with resilience and workability.

The project let us know how to better intervene with breast cancer women. Moreover, it pretends to find predictive models for the health and the quality of life of these women. This would contribute to improving the psycho-oncology units and the health system in general.

Then, the sample of both studies is not exactly the same, because the number of participants with breast cancer of the research 'Guil R. et al. Breast cancer and resilience: The controversial role of Perceived Emotional Intelligence. Front. Psychol. 2020, 11, 3604. DOI: 398 10.3389/fpsyg.2020.595713' is slightly lower than the number of participants of the present study. That is explained because the present study was done after the above-mentioned research, and in this could add new recruited women. Moreover, in the case of the sample composed of healthy women, it was not comprised of the same participants. That explains that the number of participants are not neither the same, and were other women than the other research.

  • Please check your references for example position 26 - in the journal, there is the year 2021 but you have written 2020 - check it carefully. 

Response to point 4.4.:

Dear reviewer,

We apologize for that mistake and would thank you for the comment.

In the revised version of the paper, the reference has been modified and now it is correct, changing the date from “2020” to “2021” (please, see line 495). The new reference is the following:

“Guil, R.; Gómez-Molinero, R.; Merchán-Clavellino, A.; Gil-Olarte, P. Lights and Shadows of Emotional Intelligence: Its Mediating Role in the Relationship between Negative Affect and State Anxiety in University Students. Front. Psychol. 2021, 11:615010. DOI: 10.3389/fpsyg.2020.615010”.

Reviewer 3 Report

In the present article, entitled "The Key Role of Emotional Repair and Emotional Clarity on Depression among Breast Cancer Survivors" by Guil et al., the authors aim to explore the relationship between the experience of having survived the disease and depression levels in a group of BCS and the mediating role of perceived emotional intelligence in this relationship. However, some comments need to be addressed to improve the article, its adequacy, and its readability prior to publication. My overall opinion is that this original article should be published, but only after the authors have thought about the changes below.

Introduction

  1. Line 44: "In general terms, depression seems to be more frequent in older ages, i.e., compared to the general population, Breast Cancer Survivors (BCS) older than 70 years show higher levels of depression." – The author should rewrite this paragraph and add the proper citation. As they mention, it is well known that the risk of depression symptoms is directly proportional to age, but it is mandatory to rely on data from the literature, which is sufficient on this topic.
  2. Line 45, line 77, line 88: using the abbreviations "i.e.," and "e.g.," is not mandatory and doesn’t add any asset.

Materials and methods

  1. More details need to be provided on the survey implementation. The recruitment of the participants of the 2 groups should be enlightened, and inclusion and exclusion criteria should be added.
  2. While the authors used the HADS scale, which evaluates depression and anxiety symptoms, why aren't they present in this article the anxiety sub-scale results? Did they evaluate the anxiety symptoms and decide to present them in this manuscript, or did they not add the anxiety subscale in the questionnaire? If the authors had used the anxiety sub-scale, I think using the results on anxiety would have increased the value of the scientific content of their article.
  3. Line 130: Cut-offs chosen by the authors are correct and have high sensitivity and specificity. However, depression can’t be diagnosed based on a screening scale.Depression should be diagnosed following a clinical evaluation by a professional using a variety of clinical and paraclinical evaluations.
  4. Line 142: It is important that the authors present all the socio-demographic characteristics that were assessed, besides age.

Results

  1. The authors should provide more information regarding the socio-demographic distribution of the BC group and the control group.
  2. Table 2: The authors should complete the table header with the name of the variable.
  3. Figure 1: The figure 1 is very valuable for the manuscript and contains the most important results of this study. It would be important to provide more information in the explanatory caption.

Discussion

  1. The authors should clearly describe in the discussion section what the benefits of the results of this study are in addition to the previous studies and what the results of this study contribute to. For example, how can the findings inform interventions or policy solutions?

Finally, in my opinion, a series of self-citations must be revaluated:

  1. Reference nr. 16: Ruiz-González, Paula, et al. "Resiliencia como predictora de depresión en mujeres con cáncer de mama." International Journal of Developmental and Educational Psychology 4.1 (2019): 75-84. "By contrast, other studies suggest that female survivors show better psychological adjustment and lower levels of depression than healthy women [4,16,17].", while in the quoted article, the results indicate a higher level of depression among female cancer survivors and lower resilience.
  2. Reference nr. 20: Morales-Sánchez, L.; Gil-Olarte, P.; Gómez-Molinero, G.; Guil, R. Estrategias de afrontamiento y crecimiento postraumático en mujeres, con y sin cáncer de mama. International Journal of Developmental and Educational Psychology 2019, 1(3), 95-106. "Thus, some research studies showed that one year after diagnosis, patients showed higher levels of resilience [16,19] and post-traumatic growth [20,21] than healthy women.", while in the methodology of the quoted article, there was no criteria for inclusion of one year after diagnosis.
  3. Reference nr. 25: Guil, R.; Gómez-Molinero, R.; Merchán, A.; Gil-Olarte, P.; Zayas, A. Facing anxiety, growing up. Trait emotional intelligence as 414 a mediator of the relationship between self-esteem and pre-university anxiety. Front. Psychol. 2019, 10, 567. "Also, it has been verified the importance of adequate emotional management in adaptation processes, especially when people face adverse and major changes and situations in their lives [25-27], concluding that certain emotional profiles are related to maladaptive behaviors, either in a direct or mediated way."
  4. Reference nr. 26: Guil, R.; Gómez-Molinero, R.; Merchán-Clavellino, A.; Gil-Olarte, P. Lights and Shadows of Emotional Intelligence: Its Mediat-417 ing Role in the Relationship between Negative Affect and State Anxiety in University Students Front. Psychol. 2020b, 11. "Also, it has been verified the importance of adequate emotional management in adaptation processes, especially when people face adverse and major changes and situations in their lives [25-27], concluding that certain emotional profiles are related to maladaptive behaviors, either in a direct or mediated way."

Author Response

Response to reviewer 3 comments

In the present article, entitled "The Key Role of Emotional Repair and Emotional Clarity on Depression among Breast Cancer Survivors" by Guil et al., the authors aim to explore the relationship between the experience of having survived the disease and depression levels in a group of BCS and the mediating role of perceived emotional intelligence in this relationship. However, some comments need to be addressed to improve the article, its adequacy, and its readability prior to publication. My overall opinion is that this original article should be published, but only after the authors have thought about the changes below.

Dear reviewer,

Firstly, thank you for your kindly recognition and your useful comments. The responses are the following:

Point 1: Introduction

  1. Line 44: "In general terms, depression seems to be more frequent in older ages, i.e., compared to the general population, Breast Cancer Survivors (BCS) older than 70 years show higher levels of depression." – The author should rewrite this paragraph and add the proper citation. As they mention, it is well known that the risk of depression symptoms is directly proportional to age, but it is mandatory to rely on data from the literature, which is sufficient on this topic.

Response to point 1.1:

Dear reviewer, in line with your suggestion, we have rewritten the paragraph and properly cited. The following paragraph has been added (please, see lines 44-47):

In this line, Breast Cancer Survivors (BCS) older than 70 years show higher levels of depression, compared to the general population [8,9]. This supports the studies that confirm how depression is more frequent in older ages, as stated in Girgus et al.’s study [10].”

  1. Line 45, line 77, line 88: using the abbreviations "i.e.," and "e.g.," is not mandatory and doesn’t add any asset.

Response to point 1.2.:

Dear reviewer,

The words “i.e.” and “e.g.” have been removed in the lines mentioned, as it have been recommended. Thank you for your suggestion.

Point 2: Materials and methods

  1. More details need to be provided on the survey implementation. The recruitment of the participants of the 2 groups should be enlightened, and inclusion and exclusion criteria should be added.

Response to point 2.1.:

Dear reviewer, thank you for your suggestions.

  • With regard to the survey implementation, the following information has been added in lines 136-139:

Participation was voluntary and all subjects signed previously an informed consent. In this line, all participants completed the questionnaires through an online survey by digital devices and under the supervision of research assistants. Volunteers received no financial compensation for participation in the study”.

  • Concerning the recruitment of the participants, please, see lines 125-135:

Regarding the recruitment process, the conscription of women from Group 1 (BCS) was carried out through the nursing staff of the oncology units of the three hospitals in the province of Cádiz (Spain), in coordination with the members of the research team of the project ‘PIN-0109-2018’. They explained the possibility to participate in the Project to those women who met the eligibility criteria. For those women who accepted to participate, we cite them to a personal interview to explain the details of the Project and their possibility to leave their participation at any moment, as well as to address the online questionnaire. Furthermore, the selection of women from Group 2 (healthy women) was carried out through electronic advertisements and posters in the Hospitals and Health Centers of the province. Those women who were interested to participate in the study contacted us and we also cited them as in the case of Group 1”.

  • Regarding inclusion and exclusion criteria, please see lines 108-118:

Inclusion criteria to participate in the study for both groups were: (i) to be a woman, (ii) to be older than 18 years, and (iii) to have the ability to speak and read the Spanish language. Furthermore, inclusion criteria for Group 1 (BCS) were as follows: (i) were as follows: (i) BC women with at least one year after their diagnosis or not receiving treatment, and (ii) patients with no history of cancer other than BC. 

Exclusion criteria for both groups were the following: (i) participants who could not read or write, (ii) did not have the adequate cognitive capacity to understand the instruments, (iii) presence of psychiatric or neurological disorders such as dementia or psychosis, (iv) to receive psychiatric or psychological treatment due to the presence of a mental disorder, (v) being under the influence of psychoactive medication, and (vi) having any concomitant disabling pathology.”

  1. While the authors used the HADS scale, which evaluates depression and anxiety symptoms, why aren't they present in this article the anxiety sub-scale results? Did they evaluate the anxiety symptoms and decide to present them in this manuscript, or did they not add the anxiety subscale in the questionnaire? If the authors had used the anxiety sub-scale, I think using the results on anxiety would have increased the value of the scientific content of their article.

Response to point 2.2.:

Dear reviewer,

On the one hand, the HADS scale has been implemented in its entirety, that is, both depression and anxiety have been measured. However, in the present article, we highlight only the data related to depression due to a series of limitations such as the length of the article and the number of words allowed for publication.

On the other hand, we thank you for your recommendation. The authors agree that adding the results concerning anxiety to the study would increase the scientific value of the article. However, since we must adapt to the standards of the journal and must have an adequate length, the inclusion of results referring to the anxiety variable has not been possible to be added. Nevertheless, we will take your recommendation into account and will try to publish the data obtained about anxiety, trying to relate it to other psychosocial variables.

  1. Line 130: Cut-offs chosen by the authors are correct and have high sensitivity and specificity. However, depression can’t be diagnosed based on a screening scale. Depression should be diagnosed following a clinical evaluation by a professional using a variety of clinical and paraclinical evaluations.

Response to point 2.3.:

Dear reviewer,

Thank you very much for your comment. We agree that the study would be much more complete if we compared the levels obtained in the scale with other clinical and paraclinical evaluations, which could complement our data and thus obtain more valuable results.

This aspect could be taken into account for future studies. In addition, we will add this point to the limitations of our study.

Moreover, the use of self-report measures has been also added to the limitations. Notwithstanding, the use of the most employed instruments and this type of measures such as the Hospital Anxiety and Depression Scale is needed to compare the results obtained in other studies and to increase the knowledge about the topic.

  1. Line 142: It is important that the authors present all the socio-demographic characteristics that were assessed, besides age.

Response to point 2.4.:

Dear reviewer, in addition to age, other sociodemographic variables were also measured in the study, such as marital status, employment status, educational level, having children or not, and the number of children. These will be added in the aforementioned line, and the sociodemographic data of the participants will be shown in the results. However, in the research only age will be used as a control variable in the mediation model. This aspect will be mentioned in the limitations of the study.

Therefore, the indicated paragraph has been modified in the article and the following has been added (please, see lines 176-179):

"To control the effect of age, a self-elaborated sociodemographic questionnaire was designed, which included the age of the patients. Moreover, this socio-demographic questionnaire included civil status, employment situation, level of education, having children or not, and the number of children".

Point 3: Results

  1. The authors should provide more information regarding the socio-demographic distribution of the BC group and the control group.

Response to point 3.1.:

Dear reviewer,

Thank you for your suggestion. In the section results, the characteristic of the participants has been added. Namely, the following information has been included (please, see lines 199-212):

Regarding the characteristics of the participants, the women who composed both groups presented the following aspects:

Group 1 (Breast Cancer Women): Civil status (69.6% married, 10.7% widowed, 7.1% civil, 8.9% single, 3.6% widowed); Employment situation (12.5% active, 19.6% housewife, 17.9% sick leave, 17.9% unemployed, 32.1% pensioner). Level of education (35.7% university studies, 17.9% professional training, 12.5% secondary studies, 30.4% primary studies, 3.6% no studies). On the other hand, 78.8% have children, while 21.4% have no children. Of those women with children, the mean number of children is 1.67 (M = 1.67; SD = 1.14).

Group 2 (healthy women): Civil status (50.3% married, 13.8% widowed, 8.8% civil partner, 25.4% single, 1.7% widowed). Employment situation (75.7% active, 5.0% housewife, 1.1% sick leave, 10.5% unemployed, 7.7% pensioner). Level of education (73.5% university studies, 11.6% professional training, 13.8% secondary studies, 1.1% primary studies). Of the participants, 62.4% have children, while 37.6% do not. Of those who have children, the mean number of children is 1.15 (M = 1.15; SD = 1.06).

  1. Table 2: The authors should complete the table header with the name of the variable.

Response to point 3.2.:

Dear reviewer, the header of the table has been completed with the name of the variables (please, see Table 2).

  1. Figure 1: The figure 1 is very valuable for the manuscript and contains the most important results of this study. It would be important to provide more information in the explanatory caption.

Response to point 3.3.:

Dear reviewer, following your suggestion, more information in the explanatory caption has been added. In this line, the following information has been added in the notes of Figure 1 (please, see notes of Figure 1):

Notes: The variables used in the mediation model were the following: X = Independent Variable: Breast Cancer; Y = Dependent Variable: Depression; M1 =Mediator 1: Emotional Attention; M2 = Mediator 2: Emotional Clarity; M3 = Mediator 3: Emotional Repair; Covariate = Age.

Effects: a2 = Direct effect of BC on EC; a3 = Direct effect of BC on ER; b1 = Direct effect of EA on Depression; b3 = Direct effect of ER on Depression; d21 = Direct effect of EA on EC; d32 = Direct effect of EC on ER; eM1 = Direct effect of Age on EA; c’ = Direct effect of BC on Depression; c = Total effect of BC on Depression; R2 = R square.

* = p < 0.05; ** = p < 0.01; *** = p < 0.001

Direct effects

Indirect Pathway 3 (a1b1) = Indirect effect of BC on Depression through ER only.

Indirect Pathway 6 (a2d32b3) = Indirect effect of BC on Depression through EC and ER in serial.”

Point 4: Discussion

The authors should clearly describe in the discussion section what the benefits of the results of this study are in addition to the previous studies and what the results of this study contribute to. For example, how can the findings inform interventions or policy solutions?

Response to point 4:

Dear reviewer, following your suggestions and comments, strengths and limitations has been added as the last paragraphs in section Discussion. The following information has been included in the paper (please, see lines 362-377):

Regarding the strengths of the study, results can explain the existing controversy about the presence or absence of depression in BCS and how a personal resource as PEI can act as both protector or risk factor, depending on what dimensions of PEI (EA, EC, or ER) show greater presence in the sample. In this line, we will find cases in which the different dimensions of the PEI can act as a protective factor against the development of affective disorders, reducing their incidence; or, on the contrary, as a risk factor, increasing vulnerability to experiencing negative emotional states and making difficult the adaptation after the end of treatment [23].

Therefore, the findings of the research contribute to the improvement of the health and quality of life of breast cancer survivors and, in turn, of the psycho-oncology field. In this sense, the results can be used to the development of intervention programs aimed at reducing or preventing depression in women with breast cancer. Moreover, it would let health professionals know the importance of training emotional intelligence in this population. Finally, the study highlights the need to implement policies that include emotional intelligence as a psycho-social resource that can be promoted in clinical and no clinical populations, and it would impact the health standards of the societies.

Point 5:

Finally, in my opinion, a series of self-citations must be revaluated:

  1. Reference nr. 16:Ruiz-González, Paula, et al. "Resiliencia como predictora de depresión en mujeres con cáncer de mama." International Journal of Developmental and Educational Psychology 4.1 (2019): 75-84. "By contrast, other studies suggest that female survivors show better psychological adjustment and lower levels of depression than healthy women [4,16,17].", while in the quoted article, the results indicate a higher level of depression among female cancer survivors and lower resilience.
  2. Reference nr. 20: Morales-Sánchez, L.; Gil-Olarte, P.; Gómez-Molinero, G.; Guil, R. Estrategias de afrontamiento y crecimiento postraumático en mujeres, con y sin cáncer de mama. International Journal of Developmental and Educational Psychology 2019, 1(3), 95-106. "Thus, some research studies showed that one year after diagnosis, patients showed higher levels of resilience [16,19] and post-traumatic growth [20,21] than healthy women.", while in the methodology of the quoted article, there was no criteria for inclusion of one year after diagnosis.
  3. Reference nr. 25: Guil, R.; Gómez-Molinero, R.; Merchán, A.; Gil-Olarte, P.; Zayas, A. Facing anxiety, growing up. Trait emotional intelligence as 414 a mediator of the relationship between self-esteem and pre-university anxiety. Front. Psychol. 2019, 10, 567. "Also, it has been verified the importance of adequate emotional management in adaptation processes, especially when people face adverse and major changes and situations in their lives [25-27], concluding that certain emotional profiles are related to maladaptive behaviors, either in a direct or mediated way."
  4. Reference nr. 26: Guil, R.; Gómez-Molinero, R.; Merchán-Clavellino, A.; Gil-Olarte, P. Lights and Shadows of Emotional Intelligence: Its Mediating Role in the Relationship between Negative Affect and State Anxiety in University Students Front. Psychol. 2020b, 11. "Also, it has been verified the importance of adequate emotional management in adaptation processes, especially when people face adverse and major changes and situations in their lives [25-27], concluding that certain emotional profiles are related to maladaptive behaviors, either in a direct or mediated way."

Response to point 5:

Dear reviewer,

Thank you for your indications. Reevaluating the citations that you indicated, we decided to remove the two first citations [Reference nr. 16: Ruiz-González, Paula, et al. (2019) and Reference nr. 20: Morales-Sánchez, L. et al., (2019)] due to the impact of the journal of the articles. The International Journal of Developmental and Educational Psychology Journal is not indexed in the Journal Citations Reports Systems, and they will be removed although their contents support our investigation.

Notwithstanding, we decided not to remove the last two references [Reference nr. 25: Guil, R. et al. (2019) and Reference nr. 26: Guil, R. et al. (2021)] because both studies are published in high-impact journals, and their results contribute to support our research. Both studies are aimed to prove the mediating role of perceived emotional intelligence in some psychosocial variables in other populations. Therefore, their findings increase and improve both the value of our results and of scientific research.
